# An Approach for Predicting the Low-Cycle-Fatigue Crack Initiation Life of Ultrafine-Grained Aluminum Alloy Considering Inhomogeneous Deformation and Microscale Multiaxial Strain

**DOI:** 10.3390/ma15093403

**Published:** 2022-05-09

**Authors:** Teng Sun, Lidu Qin, Yiji Xie, Zhanguang Zheng, Changji Xie, Zeng Huang

**Affiliations:** 1Key Laboratory of Disaster Prevention and Structural Safety of Ministry of Education, Guangxi Key Laboratory of Disaster Prevention and Engineering Safety, College of Civil Engineering and Architecture, Guangxi University, Nanning 535000, China; suntengqz@163.com; 2Beibu Gulf Key Laboratory of Ocean Engineering Equipment and Technology, College of Mechanical and Marine Engineering, Beibu Gulf University, Qinzhou 535011, China; 3College of Mechanical Engineering, Guangxi University, Nanning 535000, China; qinlidu@126.com (L.Q.); yuyu19971019@126.com (Y.X.); xipiao@163.com (C.X.); 173887560@163.com (Z.H.)

**Keywords:** ultrafine-grained aluminum alloy, fatigue initiation life prediction, fatigue indicator parameter, crystal plasticity

## Abstract

In this paper, a low-cycle-fatigue (LCF) crack initiation life prediction approach that explicitly distinguishes nucleation and small crack propagation regimes is presented for ultrafine-grained (UFG) aluminum alloy by introducing two fatigue indicator parameters (FIPs) at the grain level. These two characterization parameters, the deformation inhomogeneity measured by the standard deviation of the dot product of normal stress and longitudinal strain and the microscale multiaxial strain considering the non-proportional cyclic additional hardening and mean strain effect, were proposed and respectively regarded as the driving forces for fatigue nucleation and small crack propagation. Then, the nucleation and small crack propagation lives were predicted by correlating these FIPs with statistical variables and cyclic J-integrals, respectively. By constructing a microstructure-based 3D polycrystalline finite element model with a free surface, a crystal plasticity finite element-based numerical simulation was carried out to quantify FIPs and clarify the role of crystallographic anisotropy in fatigue crack initiation. The numerical results reveal the following: (1) Nucleation is prone to occur on the surface of a material as a result of it having a higher inhomogeneous deformation than the interior of the material. (2) Compared with the experimental data, the LCF initiation life of UFG 6061 aluminum alloy could be predicted using the new parameters as FIPs. (3) The predicted results confirm the importance of considering the fatigue behavior of nucleation and small crack propagation with different deformation mechanisms for improving the fatigue crack initiation life prediction accuracy.

## 1. Introduction

Aluminum alloy (AA), as a lightweight alloy, has been extensively used in structural and machine components due to its excellent mechanical properties [1]. However, with the development of industry, higher mechanical performance, such as improved hardness and strength, is required of AAs in different situations [2]. Recently, many investigations have focused on improving the mechanical properties of AAs by refining their grain size to nanoscale or UFG through severe plastic deformation techniques, among which equal channel angular pressing has become a popular strategy due to its high strength/weight ratio and promising industrial application prospects. In practice, UFG AA has been used in some key structural components of transport systems, which often endure cyclic loads [3,4]. Therefore, a thorough theoretical investigation of the fatigue properties of UFG AA is crucial in order to promote the application of such materials.

Numerous attempts have been made to identify a rational methodology for fatigue life prediction. The empirical Coffin–Manson law and its modification have been widely used by researchers to characterize the LCF of metallic materials for several decades. The Coffin–Manson law has been experimentally verified but cannot be related to the microstructure features. In order to overcome these limitations, some researchers have tried to interpret the fatigue crack initiation behavior in connection with crystal plasticity theory with the help of FIPs. Chen et al. [5] investigated the accuracy by regarding local slip accumulation and dissipated energy as FIPs to assess fatigue failure; both of these were found to correlate with the crack locations observed in experiments. The investigations conducted by Manonukul and Dunne [6] were the first to point out that the potential crack initiation location experiences the most intense accumulated plastic slip. Later, Cruzado et al. [7] argued that calculating energy dissipation in each particular slip plane is a better approach due to its clear physical background. The same conclusion was also made by Sweeney et al. [8] because the energy dissipation criterion designed in their investigations considered the working hardening induced by dislocation motion. These microstructure-sensitive FIPs identify the “hotspots” of fatigue crack initiation by judging the maximum value of the averaged FIPs, and fatigue failure is deemed to occur when the FIP computed for each cycle reaches a threshold value [7,9]. However, most of these FIPs are applied to coarse grains, which are not fully appropriate for use in UFG metals. In order to overcome these limitations, Zhang et al. [10,11,12] tried to predict the fatigue crack initiation life using a novel statistics-based criterion, providing new insights into the occurrence of fatigue failure. This research used statistical variables as FIPs to describe the inhomogeneous deformation of the representative volume element (RVE), and the fatigue life could be determined when the statistical variables reached their corresponding critical values. However, inhomogeneous plastic deformation occurring on surfaces has been overlooked in previous investigations. This topic deserves more attention [13,14], as fatigue cracks generally nucleate on the surface of UFG material.

Conventionally, local stress-based criteria for HCF and plastic strain-based criteria for LCF have been applied to estimate fatigue life by dividing the cracks into two stages, i.e., crack initiation and crack propagation [15,16]. Over the last few years, some researchers have further divided total fatigue life into three physical regimes: nucleation (10^−7^–10^−5^ m), small crack propagation (10^−5^–10^−4^ m), and long crack propagation (LC > 10^−3^ m) [17]. Additionally, the LCF fatigue crack initiation life of materials can be defined by the number of cycles required to incubate a small crack and propagate it to a macrocrack. These two stages are closely related to the microstructure topology and grain crystal anisotropy, and micromechanics-based modeling is required to accurately predict the fatigue initiation life of materials. In fact, microstructure has a dominant influence in the early stages of fatigue crack formation and growth under HCF conditions [18], while the microstructure morphology of grains/phases influences the mean (50% probability) fatigue response under LCF conditions [19]. However, most of the studies listed above do not explicitly distinguish between the nucleation and small crack propagation phases and predict the fatigue crack initiation life based solely on one FIP, which may introduce errors to the fatigue crack initiation life prediction of UFG AA.

Recently, some efforts have been made to correlate microstructural features with the early stage of fatigue crack behavior based on crystal plasticity theory. For instance, Castelluccio and Mcdowell proposed an FIP based on a modified version of the Fatemi and Socie (F–S) criterion in a high-cycle-fatigue regime [20,21,22]. In this research, the F–S criterion served as the driving force for small crack propagation. More recently, Yang et al. [23] utilized the F–S model to predict the LCF crack initiation life of GH4169 superalloy. A good prediction accuracy was achieved by assuming the small crack propagation life to account for half of the fatigue crack initiation life. In order to clarify the role of FIP in small crack propagation, a corresponding theoretical analysis and CPFEM simulations were performed by Reddy and Fatemi [18]. Their results showed that FIPs play a similar role to that of ΔK or ΔJ. However, the normal strain on the slip plane, which has been reported to not only accelerate small crack propagation [24,25] but also to lead to non-proportional cyclic additional hardening (NPCAH) [26,27], has been overlooked in previous FIP models. Meanwhile, for a crystal grain under multiaxial cyclic straining loading, the mean strain effect may play a non-negligible role in small crack propagation, as extensive macroscopic multiaxial fatigue experiments have indicated that errors could be made in small crack propagation life if the mean strain effect is not considered [28,29]. Therefore, it might be necessary to search for an FIP that is able to reflect the variation in microscale multiaxial strain, which can help us to characterize the fundamental small crack propagation mechanism.

Recently, some researchers have attempted to assess the fracture toughness of UFG metallic alloys using J-integrals. Additionally, the small crack propagation life can be predicted by associating the cyclic J-integral with the range of crack tip opening displacement. For instance, Ding and Mughrabi [30] proposed an LCF life prediction model for UFG materials based on cyclic J-integrals. In their research, the macroscale indicator parameter, i.e., the plastic strain range, is responsible for the fatigue crack growth. However, the proposed model assumes a very small initial radius of the fatigue crack, without any reference being made to the microstructure information. Similar computational models can also be applied to different kinds of UFG metals [31,32,33], but they are limited in their rendering of the heterogeneous microstructure and anisotropic mechanical behavior of materials. As outlined in [17], the assessment and prediction of early fatigue behavior (i.e., nucleation and small crack propagation ranging from 10^−7^ to 10^−5^ m) require micromechanics-based models.

In this paper, an LCF fatigue crack initiation life approach that explicitly distinguishes between the nucleation and small crack propagation regime is proposed based on experiments and crystal plasticity modeling. Choosing the UFG 6061AA as a model material, we carried out the following studies: (1) A synthetic polycrystal finite element model with a free surface was constructed and regarded as an analytical model with which to clarify the contribution of the inhomogeneous plastic deformation to the fatigue crack initiation. (2) Though conducting CPFEM simulations, the cycles required for nucleation were assessed based on the inhomogeneity of the material. (3) By introducing the FIP related to the microscale multiaxial strain field into the cyclic J-integral, the small crack propagation rate equation was obtained by associating the cyclic J-integral with the range of crack tip opening displacement. After this, the small crack propagation life was predicted based on a newly deduced Coffin–Manson model for UFG metallic alloys. (4) Comparative studies were conducted between the methods based on only one FIP and the proposed combined approach. (5) The validation of the proposed LCF crack initiation life model was carried out with respect to the errors between the predicted results and the experimental data. Finally, a technical diagram was plotted with an algorithm of all calculation steps (shown in Figure 1).

## 2. Material and Strain Fatigue Experiments

The material used in ECAP is AA 6061; its chemical composition is presented in Table 1. The process of ECAP was carried out using a wa-600b electro-hydraulic servo universal testing machine at room temperature. After eight extrusions of ECAP, the samples were machined into cylindrical specimens with a gauge diameter size of 6.25 mm and gauge length of 18 mm, as shown in Figure 2a. Meanwhile, electron backscatter diffraction (EBSD) tests were conducted to obtain the crystallographic orientation and polycrystalline geometry information prior to the LCF tests. The TSL OIM software package was employed to analyze the EBSD data. As shown in Figure 2b, the different colors in the EBSD map represent the orientations of separate grains, and the average grain size is 1.08 µm.

The fatigue test was carried out according to the American Society of Testing Materials (ASTM) E606 (Standard Practice for Strain-Controlled Fatigue Test). A Guanteng PA-20 microcomputer-controlled fatigue testing machine was used, and the gauge distance of the extensometer was 12.5 mm. Meanwhile, all the test specimens were carefully ground and polished using various grit emery papers before fatigue testing, aiming to reduce the influence of surface roughness. The LCF tests were conducted with a sinusoidal loading waveform with a frequency of 0.05 Hz at ambient temperature and carried out on three samples for reproducibility. The ECAP specimens were subjected to cyclic symmetrical tension–compression loading with a constant strain amplitude. The strain amplitudes used in the test were 0.005, 0.006, and 0.007. Finally, the stable hysteresis loops (solid lines) under different strain amplitudes were compared with the simulation results obtained from the CPFEM simulation. 

## 3. Proposed Fatigue Crack Initiation Life Prediction Approach for UFG AA

As discussed in the introduction, the LCF crack initiation life can be estimated by the superposition of fatigue nucleation life ($N_{N}$) and small crack propagation life ($N_{MSC}$), which can be written as [17]:(1)$$N_{f}=N_{N}+N_{MSC}$$

These two physically based regimes are closely related to the microstructure of polycrystalline materials. From a microstructural and physical point of view, the microscopic behavior of material influences the mean (50% probability) fatigue response in low cycle regimes [19]. Following the experimental observation of UFG AA [34,35,36], we assume that dislocation glide is still the most important deformation mechanism for UFG AA. This assumption allows us to utilize crystal plasticity theory to develop micromechanics-based fatigue criteria that relate FIPs to the phase of nucleation and transgranular small crack propagation.

### 3.1. Indicator Parameter of Inhomogeneity Considering the Influence of Normal Stress

Previous studies have demonstrated that the meso-inhomogeneous deformation of a material can be described by the standard deviation of the longitudinal strain [11,12]. For materials under LCF loading, it has been reported that a stress-assisted plastic slip-based FIP provides better predictions than one without stress terms [37]. Based on this, we proposed the $FIP_{N}$ for the nucleation regime by calculating the standard deviation of the dot product of normal stress and mesoscopic longitudinal strain (MLS) for the whole RVE, namely:

(2)$$FIP_{N}=\hat{f}$$
with
$$\hat{f}=\sqrt{\sum_{k=1}^{n_{RVE}}{\left(f\right)}_{k}^{2}p_{k}-{\left(f\right)}^{2}},\;\;\bar{f}=\sum_{k=1}^{n_{RVE}}{\left(f\right)}_{k}p_{k}$$
$$f=\sigma_{n}\cdot \varepsilon_{l\underline{l}}\;$$
$$\sigma_{n}=\left[\begin{array}{ccc} n_{x} & n_{y} & n_{z} \end{array}\right]\left[\begin{array}{ccc} \sigma_{x} & \tau_{xy} & \tau_{xz}\\ \tau_{xy} & \sigma_{y} & \tau_{yz}\\ \tau_{xz} & \tau_{yz} & \sigma_{z} \end{array}\right]\left[\begin{array}{c} n_{x}\\ n_{y}\\ n_{z} \end{array}\right]\;$$

Here, $\hat{f}$ and $\bar{f}$ represent the standard deviation and mean value of the dot product of normal stress and MLS; $n_{RVE}$ denotes the total number of finite elements; $\sigma_{n}$ is the normal stress with respect to the α slip plane, where the maximum plastic shear strain range (MSSR) occurs; and $p_{k}$ can be calculated by the volume of the *k*th element divided by the total volume of the RVE.

Furthermore, the $FIP_{N}^{Surf}$ representing the inhomogeneity on the free surface is proposed as follows:(3)$$FIP_{N}^{Surf}={\hat{f}}^{Surf}$$
with
$${\hat{f}}^{Surf}=\sqrt{\sum_{k=1}^{n_{{RVE}_{surf}}}{\left(f^{Surf}\right)}_{k}^{2}p_{k}-{\left({\bar{f}}^{Surf}\right)}^{2}},\;\;{\bar{f}}^{Surf}=\overset{n_{RVEsurf}}{\underset{k=1}{\sum }}{\left(f^{Surf}\right)}_{k}p_{k}$$
$$f^{Surf}=\sigma_{n}^{Surf}\cdot \varepsilon_{l\underline{l}}^{Surf}\;$$

It is noteworthy that all the variables calculated in Equation (3) are defined with respect to the surface layer of the RVE.

### 3.2. Modeling of Small Crack Propagation Life Based on the Indicator of Microscale Multiaxial Strain

For UFG AA, it has been observed that the micro deformation band is formed from the crack tip and moves along a close-packed plane of atoms [38]. This situation indicates that the physical modeling of small crack propagation in UFG AA should take the microscale slip mechanism into consideration. As shown in Figure 3, there exists a cyclic plastic region (CPR) on the slip plane, within which a fatigue damage region (FDR) is formed ahead of the small crack. Considering the unique deformation mechanism of UFG AA, it is not difficult to extrapolate that dynamic grain coarsening and large-scale shear-dominated deformation are supposed to occur in the FDR due to the complex dislocation interaction. In this work, the microscale control parameter used to correlate small crack propagation with the local field is characterized by FIP utilizing crystal plasticity theory.

As stated above, the local stress–strain field in the FDR evolves in a multiaxial and non-proportional manner, giving rise to NPCAH behavior in the realistic microstructure. Previous macroscale experiments have proven [39] that NPCAH behavior can be described by establishing a multiaxial fatigue damage parameter with respect to the normal strain and maximum shear strain based on the critical plane method. Therefore, based on the assumption that only active slip planes can be potentially considered as critical planes [20], a multiaxial fatigue damage parameter, defined on the specific crystallographic plane, is proposed to reflect the NPCAH effect at the grain level, which can be deduced according to the von Mises yield criterion:(4)$$\frac{\Delta \varepsilon_{eq,N}}{2}={\left[{\left(\varepsilon_{n}^{*}\right)}^{2}+\frac{1}{3}{\left(\frac{\Delta \gamma_{\max}}{2}\right)}^{2}\right]}^{\frac{1}{2}}$$
where $\Delta \gamma_{\max}$ is the MSSR in the α slip system and $\varepsilon_{n}^{*}$ denotes the normal strain on the crystallographic plane where the MSSR occurs; it can be calculated as $\varepsilon_{n}^{*}=\underset{t_{A}<t<t_{E}}{\max}(\varepsilon_{n}\left(t\right))-\underset{t_{A}<t<t_{E}}{\min}(\varepsilon_{n}\left(t\right))=\varepsilon_{n}^{\max}-\varepsilon_{n}^{\min}$. $\varepsilon_{n}$, representing the normal strain on the slip plane, can be defined as:(5)$$\varepsilon_{n}^{\left(\alpha \right)}=\left[\begin{array}{ccc} n_{x} & n_{y} & n_{z} \end{array}\right]\left[\begin{array}{ccc} \varepsilon_{x} & \gamma_{xy} & \gamma_{xz}\\ \gamma_{xy} & \varepsilon_{y} & \gamma_{yz}\\ \gamma_{xz} & \gamma_{yz} & \varepsilon_{z} \end{array}\right]\left[\begin{array}{c} n_{x}\\ n_{y}\\ n_{z} \end{array}\right]$$

Considering that the mean range on the specified slip plane may play an important role in the small crack growth, the equivalent mean strain range, combined with the multiaxial fatigue damage parameter, is regarded as the local driving force in small crack propagation. Consequently, the new proposed indicator parameter of the microscale multiaxial strain field can be rewritten as:(6)$$FIP_{P}=\frac{\Delta \varepsilon_{eq,N}}{2}+\frac{\Delta \varepsilon_{eq,m}}{2}$$
with
$$\begin{array}{l} \frac{\Delta \varepsilon_{eq,m}}{2}=\mathrm{sgn}\left[\mathrm{sgn}(\frac{\Delta \varepsilon_{n,m}}{2})\cdot {\left(\frac{\Delta \varepsilon_{n,m}}{2}\right)}^{2}+\frac{1}{3}\mathrm{sgn}(\frac{\Delta \gamma_{m}}{2})\cdot \mathrm{sgn}{\left(\frac{\Delta \gamma_{m}}{2}\right)}^{2}\right]\\ \;\;\;\;\;\;\;\;\;\;\;\;\;\times {\left[{\left(\frac{\Delta \varepsilon_{n,m}}{2}\right)}^{2}+\frac{1}{3}{\left(\frac{\Delta \gamma_{m}}{2}\right)}^{2}\right]}^{\frac{1}{2}} \end{array}$$
where $\Delta \varepsilon_{n}^{m}$ and $\Delta \gamma^{m}$ represent the mean normal strain range and mean shear strain range for the slip plane, respectively.

The relationship between the cyclic J-integral and FIP was clarified in the work of McDowell [40]. For UFG materials, it has been recognized that the cyclic J-integral shows a good correlation with the fatigue crack growth rate under the elastic–plastic field for the case of an LCF condition, and it was suggested as an effective alternative to the application of ΔK in the investigations for both short crack and long crack growth regimes [41,42]. In this paper, the cyclic J-integral is proposed to be responsible for the small crack propagation, and it follows a representative relationship based on the work of Rice [43]:(7)$$J=-\frac{\partial V_{int}}{\partial r}$$

Here, *V* denotes the internal energy dissipated in the process of small crack propagation. Previous studies have already demonstrated that the growth rate of fatigue cracks for UFG metals depends on whether the local cyclic peak stress in the fatigue degradation zone ahead of the crack tip approaches the ultimate tensile strength [44,45,46]. Therefore, the interaction energy dissipated per unit length can be determined using:(8)$$V_{\mathrm{int}}=-R_{Y,UFG}{\tilde{\varepsilon }}_{FDR}\pi {\left(r_{FDR}/2\right)}^{2}$$

Here, $R_{Y,UFG}$ denotes the cyclic ultimate tensile strength and ${\tilde{\varepsilon }}_{FDR}$ represents the accumulated mean multiaxial damage strain in FDR, which can be calculated by the integration of the strain in CPR ($\varepsilon_{CPZ}$) over the radius of the fatigue damage region ($r_{FDR}$) [33,47,48]. Then, the mean multiaxial damage strain can be obtained by the integral equations as follows:(9)$${\tilde{\varepsilon }}_{FDR}=\frac{1}{r_{FDR}}\int_{0}^{r_{FDR}}\varepsilon_{CPR}(r)dr$$

In order to consider the effect of multiaxial damage strain on the small crack propagation, we extend the model in [30,49] by incorporating the $FIP_{p}$ into the formulation of $\varepsilon_{CPR}$ and $r_{FDR}$. The newly developed model is described by:(10)$$\varepsilon_{CPR}(r)=\frac{FIP_{P}}{2}{\left(\frac{r_{c}}{r}\right)}^{1/(n^{\prime }+1)}$$
(11)$$r_{FDR}=\frac{\lambda \pi K_{GBS}^{2}}{16K_{GBC}^{2}}\left(\frac{{\left({K^{\prime }}_{UFG}\right)}^{3+1/n^{\prime }}}{R_{m,UFG}^{1+1/n^{\prime }}R_{Y,UFG}^{2}}\right){\left(\frac{FIP_{P}}{2}\right)}^{3n^{\prime }+1}\alpha$$

Here, λ is the cyclic plastic zone correction factor, which can be obtained by one experimental fatigue test; ${K^{\prime }}_{UFG}$ and $n^{\prime }$ are material parameters related to the cyclic mechanical response of UFG AA; $K_{GBS}$ and $K_{GBC}$ are, respectively, the GB strengthening (GBS) factor and the GB constraint (GBC) factor. Additionally, both of these can be calculated from the relationship between the cyclic mechanical response and the GB properties of UFG materials [30]
(12)$$\{\begin{array}{c} K_{GBS}\equiv R_{Y,0.2}^{UFG}/R_{Y,0.2}\\ K_{GBC}\equiv [(\Delta \sigma /2)/(\Delta \sigma_{eff}/2)]/2 \end{array}$$
where $R_{P,UFG}$ and $R_{P}$ represent the effective yield strength of UFG and its CG counterparts and $\Delta \sigma /2,\Delta \sigma_{eff}/2$ are, respectively, the cyclic stress range and equivalent stress range at the steady cycle, which can be obtained by performing CPFEM simulation. These parameters reflect the strengthening effect on the metal matrix and the constraint on the dislocation motion due to the large amount of grain boundaries (GBs) in UFG materials.

After some computation, one can further obtain Equation (13) by substituting Equation (8) into Equation (7), namely:(13)$$\Delta J=\frac{\lambda \pi^{2}K_{GBS}^{2}}{32K_{GBC}^{2}}(\frac{n^{\prime }+1}{n^{\prime }})\left(\frac{{\left({K^{\prime }}_{UFG}\right)}^{3}}{R_{p,UFG}^{2}}\right){\left(\frac{FIP_{P}}{2}\right)}^{3n^{\prime }+1}\alpha$$

Based on the fracture mechanics, the ΔJ parameter and the range of crack tip opening displacement (ΔCTOD) are assumed to satisfy the following relation [50]:(14)$$\Delta CTOD=\Delta J/\omega R_{P,UFG},\;with\;\omega =1.5$$

Therefore, with the knowledge of Equations (13) and (14), we obtain the fatigue crack growth rate equation for the UFG material as:(15)$${\left(\frac{da}{dN}\right)}_{MSC}=\frac{\lambda \pi^{2}K_{GBS}^{2}}{96K_{GBC}^{2}}(\frac{n^{\prime }+1}{n^{\prime }}){\left(\frac{{K^{\prime }}_{UFG}}{R_{p,UFG}}\right)}^{3}{\left(\frac{FIP_{P}}{2}\right)}^{3n^{\prime }+1}a$$

Since the cracks will evolve with the loading cycles and finally reach a critical value, one can obtain the fatigue life formula by integrating Equation (15) from an assumed small crack $r_{i}$ to a critical crack size $r_{f}$. The expression of FCP life for the considered material is then:(16)$$N_{f}=\left[\frac{96K_{GBC}^{2}}{\lambda \pi^{2}K_{GBS}^{2}}\left(\frac{n^{\prime }}{n^{\prime }+1}\right){\left(\frac{R_{p,UFG}}{K_{UFG}^{\prime }}\right)}^{3}\ln\left(\frac{r_{f}}{r_{i}}\right)\right]{\left(\frac{FIP_{P}}{2}\right)}^{-\left(3n^{\prime }+1\right)}$$

After rearranging the equation above, one can find that Equation (16) can be easily translated into the standard Coffin–Manson model, i.e., Equation (17), with the consideration of the deformation mechanism at the microscale and its corresponding mechanical response at the macroscale:(17)$$\frac{FIP_{P}}{2}={\left[\frac{192K_{GBC}^{2}}{\lambda \pi^{2}K_{GBS}^{2}}\left(\frac{n^{\prime }}{n^{\prime }+1}\right){\left(\frac{R_{p,UFG}}{K_{UFG}^{\prime }}\right)}^{3}\ln\left(\frac{r_{f}}{r_{i}}\right)\right]}^{\frac{1}{3n^{\prime }+1}}{\left(2N_{f}\right)}^{-1/\left(3n^{\prime }+1\right)}$$

## 4. Simulation Methodology

### 4.1. Dislocation-Based Constitutive Model

The following formulations of the dislocation-based constitutive model for UFG material were based on the theory established in [51]. In order to explore the cyclic plasticity behavior of UFG AA, the flow rule was modified by introducing a nonlinear kinematic hardening term to the slip system, i.e.,
(18)$${\dot{\gamma }}^{\left(\alpha \right)}=\upsilon_{0}\frac{b}{d}\exp\left\{-\frac{\Delta G_{0}}{KT}{\left[1-{\langle \frac{\left|\tau^{\left(\alpha \right)}-\chi^{\left(\alpha \right)}\right|-\tau_{ath}^{\left(\alpha \right)}}{{\hat{\tau }}_{UFG}}\rangle }^{p}\right]}^{q}\right\}\mathrm{sgn}(\tau^{\left(\alpha \right)}-\chi^{\left(\alpha \right)})$$

Here, the brackets ⟨…⟩ denote the Macaulay bracket, meaning that ⟨x⟩ ≡ x for x ≥ 0; otherwise ⟨x⟩ ≡ 0. *P* and *q* are two material parameters. $\upsilon_{0}$ denotes the attempt frequency and *b* and *d* represent the Burgers vector and the mean grain size of UFG metal, respectively. The physical meaning of “*b*/*d*” corresponds to the plastic shear strain generated in the progress of dislocation sliding from GBs to GBs. $\Delta G^{\left(\alpha \right)}$ denotes the reduced activation energy; α refers to the total amount of slip in the system, which is equal to 12 for FCC polycrystalline metal; ${\dot{\gamma }}^{\left(a\right)}$ is the inelastic strain rate for the active a-slip systems, i.e., a {110} <110> system for an FCC crystal structure; and $\tau^{\left(\alpha \right)}$, representing the resolved shear stress, can be computed as:(19)$$\tau^{\left(\alpha \right)}=M^{e}:P^{\left(a\right)},\;with\;\;M^{e}=F^{eT}F^{e}\cdot T^{e}$$

Here, $M^{e}$ denotes the Mandel stress tensor; $T^{e}$ represents the second-order Piola–Kirchhoff stress tensor; $E_{e}$ is the elastic Green–Lagrange strain tensor; and C is the fourth-order elastic tensor. Based on the Hooke’s law, their relationship can be written as:(20)$$T^{e}=C:E^{e}\;\mathrm{with}\;E^{e}=\frac{1}{2}\left(F^{eT}F^{e}-I\right)$$

The lattice friction stress, ${\hat{\tau }}_{\mathrm{UFG}}$, is the critical shear stress used to generate crystal plastic slip at 0 K, which can be obtained based on a previous investigation into UFG metals [51]:(21)$$\hat{\tau }=2\Delta G_{0}/Lb\omega_{0}$$
where $\omega_{0}$ represents the average travel distance of the dislocation and L represents the distance of two pinning obstacles, which can be determined from the product of volume fraction (c) and grain size as L=md (0<m<1).

To account for the evolution of back-stress at the grain level, a kinematic hardening rule including a strain hardening term and a dynamic recovery term is introduced based on the work of Busso and McClintock [52], namely:(22)$$\chi^{\left(\alpha \right)}=h_{b}{\dot{\gamma }}^{\left(\alpha \right)}-\frac{r_{D}}{\tau_{ath}^{\left(\alpha \right)}}\chi^{\left(\alpha \right)}\left|{\dot{\gamma }}^{\left(\alpha \right)}\right|$$

Here, $h_{b}$ and $r_{D}$ are two material parameters and $\tau_{ath}^{\left(\alpha \right)}$ represents the slip resistance law in the α slip system [53]:(23)$$\tau_{ath}^{\left(\alpha \right)}=\mu Gb\sqrt{\sum_{\beta =1}^{12}A^{\alpha \beta }\rho^{\beta }},\;\;\;\;\;A^{\alpha \beta }=h_{0}\left[\omega_{1}+(1-\omega_{2})\sigma^{\alpha \beta }\right]$$
where μ is a material parameter that is equal to 0.7, G is the shear modulus, and $h_{0}$ is a scalar function describing the strength of dislocation pair interaction. Generally, $A_{\alpha \beta }$ is a 12 × 12 dislocation interaction matrix that describes the degree of obstruction between different slip systems. In this investigation, an interaction matrix consisting of two elements corresponding to weak interactions and strong interactions is adopted for simplicity.

Considering the competition between the multiplication and annihilation of dislocations for the α slip system, the overall dislocation density evolution equation is governed by the balance law, which is suggested as:(24)$${\dot{\rho }}^{\left(\alpha \right)}=\frac{\left|{\dot{\gamma }}^{\left(\alpha \right)}\right|\;}{b}\left(\frac{1}{d}+\frac{\sqrt{\sum_{a\neq \beta }\rho^{\beta }}}{K}-y_{k}\exp(-\frac{\Delta GB}{k_{B}T})\rho^{\left(\alpha \right)}\right)\;$$
where *K* is a material constant that controls the size of the mean free path [54]. For UFG materials, the process of the absorption of dislocations by GBs is proportional to the diffusivity along the GBs over the course of cyclic loading. Therefore, the absorption rate of dislocations at GBs is proposed to be related to the temperature and thermal activation energy of GBs.

### 4.2. Construction of the Finite Element Model

#### 4.2.1. Applied Boundary Conditions

The loading direction applied to the polycrystalline finite element model is shown in Figure 4a. Additionally, the specific boundary conditions can be implemented using the following steps: (1) Normal displacement was applied with U3 = U for the positive surface of the third axis, while U3 = 0 was applied for the negative surface of the third axis. (2) On the negative surface of axes 1 and 2, the displacement in the normal direction of all nodes was equal to zero. (3) For the positive surface of the first axis, node A at the lower right corner of the surface was taken as the reference point, while the other nodes on the surface were recorded as A′ set. In order to meet the condition of macro-uniaxial loading and continuous material deformation, the constraint Equation (25) was embedded into ABAQUS to satisfy the boundary conditions above.
(25)$$-U_{1}^{A}{+U}_{1}^{A^{\prime }}=0$$

#### 4.2.2. Validation of the RVE

Based on the microscopic experiment of UFG 6061AA, it is acknowledged that Goss texture is formed for UFG 6061AA in the process of ECAP. In other words, the polycrystalline aggregate will show obvious anisotropy. Therefore, the number of grains in RVE should be sufficient to reflect the plasticity heterogeneities of the real microstructure. However, numerical simulations with a large number of grains under a crystal plasticity framework are very costly from a computational viewpoint. Thus, a reasonable number of grains with the reduced crystallographic texture should be used for the accurate prediction of mechanical response at the polycrystal level. In this investigation, four RVEs with the same number of elements (27,000) but containing 27, 64, 216, and 512 grains (Figure 4c–f, different colors represent different grains) were constructed by grain size analysis (as shown in Figure 4b) and assigned a reduced crystallographic texture extracted from the pole map (Figure 5). The computationally obtained results and their local enlarged drawing at the peak stress are plotted in Figure 6. The difference in macro hysteresis loop shows that the peak stress identified for 216 grains is slightly higher than that for 512 grains, but less than 2 MPa. Therefore, a 3D polycrystalline RVE established using 27,000 elements and 216 grains was adopted in the following CPFE simulations.

### 4.3. Parameter Identification

#### 4.3.1. Identification of Material Parameters and Simulating Cyclic Deformation Behavior

The material parameters used in the crystal plasticity constitutive equation were classified into three categories. The first category contained the microstructural features of UFG metals, including grain size and grain orientation, which were consistent with those observed in the EBSD experiment. The second category was mainly physical material parameters, such as the GB thermal activation energy, Boltzmann constant, and Burger vector, all of which were related to the unique properties of UFG metallic material. These parameters controlling the development of isotropic and kinematic hardening were initialized based on previously published studies on UFG AA at room temperature [47,55,56,57]. Lastly, the remaining parameters, such as the three independent single-crystal elastic constants and dynamic recovery rate, were adjusted from those in the symmetric strain-controlled test using a “trial-and-error” method. All the calibrated parameters discussed above are listed in Table 2.

Finally, the rationality of the material parameters was verified in terms of the stable stress–strain hysteresis loops and their corresponding numerical simulation. Figure 7 shows a comparative analysis between the simulated stable stress–strain hysteresis loops and the experimental ones at the strain amplitudes of 0.5%, 0.6%, and 0.7%. The maximum stress difference between the test data and the micro-level simulations is less than 3 MPa, which indicates that the obtained simulated stable hysteresis curves are in good agreement with the experimental results. Meanwhile, all the simulated results predict a slight cyclic softening, which is consistent with the experimental observation. Moreover, in order to reflect the reduction in dislocation density observed in the cyclic loading process, two internal variables, i.e., initial dislocation density and critical shear stress, are reduced with the microstructure evolution. In summary, all the above results show that the calibrated material parameters for the crystal plasticity model are reasonable and acceptable for use in further investigations.

#### 4.3.2. Identification of Material Parameters for Small Crack Propagation Equation

The small crack propagation life of UFG 6061AA was predicted according to Equation (16). Obviously, the main factors that affect the small crack propagation life of UFG 6061AA are the defined initial fatigue crack length, the material parameters, and the investigated material. Thus, a series of tension–compression symmetric strain-controlled experiments were conducted carefully to identify these key parameters prior to the application of UFG 6061AA. However, very limited information about the fatigue crack initiation life of UFG AA is available in the literature. Additionally, it is hard to measure the corresponding real-time crack length of a smooth cylindrical specimen using the currently available observation equipment. Therefore, in this work, an empirical approach outlined in the ASTM E606 standard was adopted; i.e., the total fatigue crack initiation life was estimated when the maximum stress of the cyclic stress–strain curve had dropped below 5% of the stabilized hysteresis loop peaks. Then, the critical crack size was determined based on the fracture appearance using SEM observation. In this work, the critical crack size for UFG 6061AA was taken as 25 μm, which is approximately 20 times larger than the initial mean grain size. In order to obtain the material parameters related to the cyclic plasticity behavior, a set of cyclic stress–strain curves for UFG and CG 6061AA were plotted based on a series of symmetrical cyclic strain-controlled experiments (shown in Figure 8). It should be noted that the cyclic yield strength was calculated by the formulation fitted by the approximation $R_{P,0.2}=K^{\prime }{\left(0.002\right)}^{n}$ [30]. In addition, the GBS factor can be obtained by the experimental method reported in [58] or the CPFEM simulation. Finally, all the parameters used in our calculation are listed in Table 3.

## 5. Results and Discussion

### 5.1. Determining the Statistical Variable Describing the Inhomogeneity

According to the previous experimental observations of fatigue experiments, micro-cracks tend to nucleate on the surface of metal rather than in the bulk [59]. The reason for this is that the surface of samples generally suffers from manufacturing defects and a poor finish, leading to the accumulation of damage that accelerates the fatigue failure of components under cyclic loading. Based on the use of an RVE with preferred orientation according to the EBSD, Figure 9a,b compare the distribution of mesoscale longitudinal strain at the maximum tension point of the 3rd and 300th cycles, respectively. It can be seen that the free surface becomes obviously uneven as the number of cycles increases. Aiming to quantitatively depict the degree of deformation of the RVE and its free surface, the inhomogeneity of the polycrystalline aggregation and its free surface are described by Equations (2) and (3), respectively.

Figure 10 displays the histograms of the frequency distribution for f and $f^{Surf}$ under various strain amplitudes. It is evident that the statistical distributions for f and $f^{Surf}$ are Gaussian-like, similar to those reported in [11,12]. Therefore, it is reasonable to adopt these two parameters to signify the level of dispersity of the inhomogeneous local strain field. As shown in Figure 10a–c, the SD of f presents a trend of growth, which implies that the inhomogeneity of the material increases with the increase in the applied strain amplitude. Figure 10c,d compare the SD of f for the whole RVE at an amplitude of 0.7%, and it is found that the SD of f at the 300th cycle is 20 times larger than that at the 3rd cycle. Furthermore, Figure 10e,f investigate the inhomogeneity on the free surface of the RVE by calculating the $f^{Surf}$ at the 300th and 3rd cycle with respect to the strain amplitude of 0.7%. A very similar evolution was found but with a 28.25 times difference. When comparing Figure 10c,d with e,f, the SD of $f^{Surf}$ is obviously higher than that of f in the 300th and 3rd cycle. Based on the analysis above, we can conclude the following: (1) The cycle numbers and the applied loading conditions are the two key external factors that affect the inhomogeneity of materials. (2) Even if there is no damage on the surface, the metal surface suffers from more inhomogeneous plastic deformation than the interior of the material, which further clarifies the contribution of the plasticity heterogeneities and micromechanical interactions between neighbor grains to the fatigue damage mechanism. (3) The SD of $f^{Surf}$ calculated with respect to the free surface layer of the RVE is more suitable for use as a fatigue criterion to judge the fatigue nucleation occurrence.

### 5.2. Analysis of Local Stress and Strain Field on the Free Surface

From the microscopic viewpoint, the inhomogeneity of polycrystalline aggregation can be described by the Schmid factor of individual grains. The contours of the Mises stress and maximum principal strain are plotted in Figure 11; they were calculated by ABAQUS after computational stabilization at the 10th cycle with respect to the tension peak. The heterogeneous state of plastic deformation at the grain level was mainly caused by the preferred orientation processed by ECAP. Since the condition of stress equilibrium and strain compatibility needs to be satisfied during deformation, the misorientation of the neighboring grains makes the GBs “hot ports” in the polycrystalline aggregation. As displayed in Figure 11a,b, more details of the local stress and strain response on the free surface of the RVE were investigated along two horizontal lines. Moreover, two types of grain combinations, that is to say, the hard–soft combination and the soft–soft combination, were selected to analyze the effect of crystal orientation on the inhomogeneous local stress and strain field. It can be observed from Figure 12a,b that the maximum Mises stress appears at hard grains, while the minimum value appears at soft grains. On the other hand, it can be seen that the transition process of stress and strain between two soft grains tends to be relatively smooth. For the two adjacent soft grains, the elastic and plastic behavior at the micro-level is anisotropic. Therefore, it can be inferred that this inhomogeneous deformation process firstly induces a grain shape change to satisfy the strain compatibility and then stress equilibrium along the GBs. Finally, it should be noted that the combination of hard grain and hard grain was been discussed in this investigation, but a similar evolution could be inferred from the evolution law of the soft–soft combination.

In order to further validate the aforementioned deduction, two soft grains and one hard grain were selected to explore the microscale cyclic plastic response at the grain level. Figure 12c shows the mechanical responses of individual grains obtained from soft grains 67 and 70 and hard grain 169 with respect to the 10th cycle. It can be seen clearly that the two soft grains exhibit a higher strain response and a lower stress peak as compared to the hysteresis loops obtained from the experiment, while the hard grain exhibits a higher stress response and a lower strain response. These results further demonstrate that the material properties of hard grains and soft grains are anisotropic in terms of elastic and plastic deformation, which are, respectively, linked to elastic lattice distortion and the plastic slip accumulated over the course of cyclic loading. Moreover, one can see that the yield strength, Young’s modulus, and cyclic response for the soft grains are smaller than those obtained from the experiments, which is in contrast to the situation seen in hard grains. These well-oriented grains, which are prone to slip, will contribute to a larger accumulation of micro-level plastic deformation and higher hysteresis energy dissipation.

### 5.3. Prediction of Low-Cycle-Fatigue Crack Initiation Life

#### 5.3.1. Nucleation Life Based on the Evolutions of the Inhomogeneity on the Free Surface

Further investigations of the relationship between the inhomogeneous meso-plastic deformation and fatigue nucleation were conducted by tracking the evolution of $FIP_{N}^{Surf}$ through CPFEM simulation. For UFG AA, the inclusion of large amounts of precipitates distributed dispersedly within the matrix was observed; this contributes to the localized crystallographic slip concentration, accelerating the nano-void nucleation and growth [35,37] and resulting in the scatter characteristic of nucleation life in UFG materials. At present, there are technical difficulties in tracking the real-time evolution of nucleation and small cracks in smooth cylindrical fatigue specimens using the currently available observation equipment. Therefore, we made the assumption that the fatigue nucleation regime may account for 30%, 50%, 70%, and 100% of the fatigue initiation life for simplicity.

Figure 13 shows the evolution curves of the simulated $FIP_{N}^{Surf}$, characterizing the inhomogeneous micro-plastic deformation that has occurred under three different strain amplitudes in four probabilities as the number of cycles increases. It can clearly be seen that the $FIP_{N}^{Surf}$ increases as the number of cycles increases, and the evolution law is similar within the four assumed probability ranges. The simulation results also indicate that the inhomogeneous deformation on the free surface became increasingly serious during the cyclic loading. The relationship between the experimental data and their averaged values is marked by red solid dots and blue crosses distributed along the FIP evolution curves. The variation ranges of $FIP_{N}^{Surf}$, i.e., 2.6∼15.8, 6.7∼28, 9∼40.2, and 14∼57.8, were divided into six equal parts by seven critical values. Subsequently, a group of horizontal lines, including four black dotted lines and three solid lines plotted in red, pink, and green, was drawn though these critical values. The values of $FIP_{N}^{Surf}$ corresponding to the intersections of these critical lines and FIP evolution curves are listed in Table 4.

Selecting the limiting values in Table 4, the comparisons between the experiment life and the predicted nucleation life are shown in Figure 14. The abscissa is the nucleation life calculated from the product of the total fatigue initiation life and the assumed probability. The ordinate is the predicted nucleation life obtained by utilizing the proposed $FIP_{N}^{Surf}$. The bold diagonal solid orange line represents perfect fatigue crack initiation life prediction. Compared with Figure 14a,b, it can be seen that all the predicted data points are within the ±3.0 error band. However, compared with the experimental results, this approach overestimates the nucleation life at larger values of strain amplitude, while showing conservative predicted results at lower values of strain amplitude. The reason for the lower predictability of this model can be attributed to the fact that it exclusively focuses on the inhomogeneous plastic deformation of the material, but ignores the effect of the inclusion of precipitates dispersed in large amounts of UFG aluminum alloy matrix, which may trigger early micro-crack nucleation due to deformation incompatibility. This causes the nucleation life to be lower than expected in the case of higher strain amplitudes. On the other hand, the predicted results tend to be conservative under lower strain amplitudes. This is mainly due to the smaller grain size in the multiaxis stress state being conducive to more homogeneous slipping plastic deformation in the UFG regime. As a result, the stress concentration is reduced at the micro level, which delays the propagation of small cracks to macroscopic ones [60,61]. This is manifested as an improvement in the fatigue strength of UFG metals at the macro level. In addition, it is necessary to point out that the statistical predicted method based on inhomogeneity can better capture the scatter characteristic of nucleation behavior to an extent. Although the approach based on inhomogeneous plastic deformation is capable of assessing the fatigue initiation life of UFG metallic alloys, more thorough investigations that take into particular consideration the unique microstructure characteristics and the macroscopic mechanical response of UFG material are required to establish a more physically based equation for this kind of material.

#### 5.3.2. Small Crack Propagation Life Prediction Based on Microscale Multiaxial Strain

Figure 15a shows the distributions of the $FIP_{P}$ for 207 grain sets in the RVE by contours at a strain amplitude of 0.005 after numerical cyclic stability. The results show the $FIP_{P}$ presents a strong inhomogeneous nature with values varying in the range of −6.614 × 10^−7^ to 1.535 × 10^−2^. In order to characterize the $FIP_{P}$ with respect to the strain amplitude of 0.5%, 0.6%, and 0.7%, the maximum value of $FIP_{P}$ was extracted from the corresponding grain sets after cyclic deformation. The data points obtained are displayed in the scatter diagram shown in Figure 15b–d. It can be seen the maximum $FIP_{P}$ for the crystal plasticity model obtained under different strain amplitudes varies in the range of 2.38 × 10^−3^ to 1.54 × 10^−2^, 3.58 × 10^−3^ to 1.84 × 10^−2^, and 4.11 × 10^−3^ to 2.14 × 10^−2^, respectively. The homogenized values of the largest $FIP_{MSC}$ are, respectively, 7.46 × 10^−3^, 8.94 × 10^−3^, and 1.05 × 10^−2^. Subsequently, these homogenized $FIP_{P}s$ are introduced into Equation (16) to calculate the small crack propagation life. Furthermore, through the analysis of the distribution of the maximum $FIP_{P}$ value of crystal sets, it can be determined that the stress–strain states at the critical sites evolve in a multiaxial and non-proportional manner. From the data shown in Figure 16, the errors between the predicted results and the experimental data can be seen to be acceptable when the error band is set to the range of ±3.0. It is noteworthy that the range of four probabilities corresponds to different cyclic plastic zone correction factors for different deformation mechanisms. In addition, it is not difficult to find that the fatigue initiation life approach based on the microscale multiaxial strain, corresponding to 100% probability, is obviously conservative as a result of negligence in the nucleation regime.

#### 5.3.3. Fatigue Crack Initiation Life Prediction Based on the Combination of Inhomogeneity and Microscale Multiaxial Strain

As shown in Figure 17, the fatigue crack initiation life predicted by the proposed methodology is more reasonable than that based on only one FIP (corresponding to 100% probability) and the previously published method [61] under various macroscopic strain amplitudes. When the percentage of nucleation regime varies from 50% to 70% probability, all the predicted data points are within ±1.5 error bands. The best prediction accuracy was achieved when the phase of nucleation accounted for 70% of the LCF crack initiation life. The prediction data and their errors with the averaged fatigue lives for different prediction models with respect to the corresponding half strain range, $\Delta E_{t}/2=0.5\%,\;0.6\%,\;0.7\%$, are also listed in Table 5 and Table 6. To describe the relationship between the probability of FN, applied strain amplitude, and fatigue initiation life, a three-dimensional fatigue initiation life histogram was drawn, as shown in Figure 18. It can be clearly seen that the fatigue crack initiation life increases along with the probability of FN, while decreasing as the applied strain amplitude increases. Meanwhile, it can also be seen that the fatigue initiation life is dominated by the applied strain amplitude. The findings shown above further indicate that the influences of the inhomogeneous plastic strain for fatigue initiation life are significant at lower applied strain ranges, while the influences of the multiaxial strain field for small crack propagation are remarkable only at larger strain ranges. The proposed method, which explicitly considers the phase of nucleation and small cracks, can be used to provide an accurate description of the early behavior of UFG metallic alloys through CPFEM simulation. Since our approach considers the heterogeneity characteristics arising from crystallographic orientation and the microscale LCF multiaxial strain field, it can capture the grain-scale deformation localization well and therefore improve the precision of fatigue crack initiation life prediction.

## 6. Conclusions

In this work, an LCF crack initiation life prediction approach was proposed that explicitly distinguishes the nucleation and small crack propagation regime; it contains two newly proposed FIPs relating to inhomogeneity and the microscale multiaxial strain field. Micro-level numerical simulations and a series of strain-controlled fatigue experiments were conducted for UFG 6061AA in order to examine the prediction capacities of the proposed model. The main findings of this study can be summarized as follows:

(1)By performing a statistical analysis of the RVE and its free surface, it was proven that, even if there are no machining defects or damage on the free surface of the specimen, this is still the most dangerous place for fatigue nucleation to occur due to the evolution of inhomogeneous deformation. Additionally, stress-assisted FIP based on the statistical method is capable of expressing the degree of inhomogeneity of UFG material.(2)Regarding the two newly proposed FIPs as the driving force for the nucleation and small crack propagation, we predicted the nucleation life and small crack propagation life with respect to different probability factors. The predicted accuracy of the fatigue crack initiation life based on only one FIP was acceptable when the error bands were set in the range of ±3.(3)When the phase of nucleation accounted for 50% to 70% of the LCF crack initiation life for three different strain amplitudes, the predicted accuracy of the developed numerical process was improved, with all the predicted data points lying within the ±1.5 error band. The proposed methodology accompanied by the two FIPs provides new insights into the early stage of the LCF fatigue behavior of UFG AA.

However, the criterion used to judge fatigue failure may warrant further research to take into account the effect of the surface roughness and material internal defects in engineering materials and structural components. Moreover, in situ experimental observation will be conducted in future work to better understand the LCF properties of UFG aluminum alloy and provide a more accurate verification of the proposed methods.

## Figures and Tables

**Figure 1 materials-15-03403-f001:**
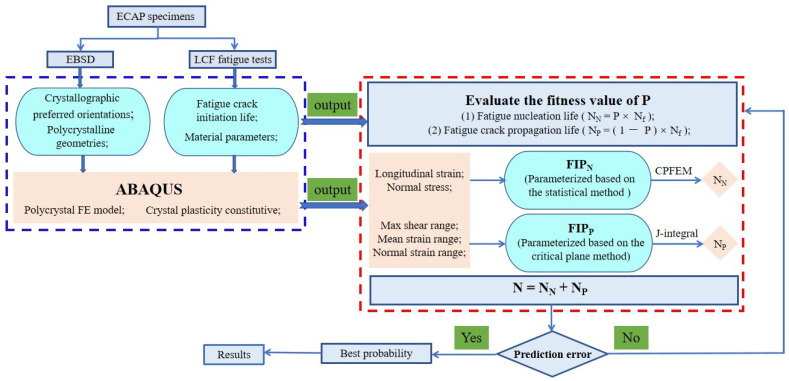
The technical diagram for the proposed method.

**Figure 2 materials-15-03403-f002:**
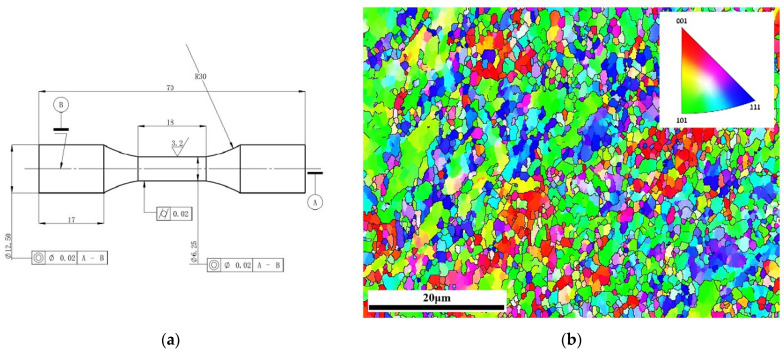
(**a**) Geometric dimensions of the cylindrical specimen (unit: mm); (**b**) EBSD map of UFG AA6061.

**Figure 3 materials-15-03403-f003:**
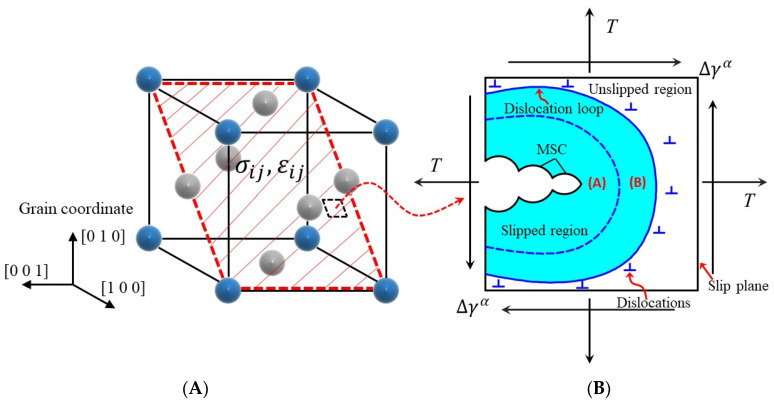
Propagation of microstructurally small crack: (**A**) fatigue damage region, (**B**) cyclic plastic region.

**Figure 4 materials-15-03403-f004:**
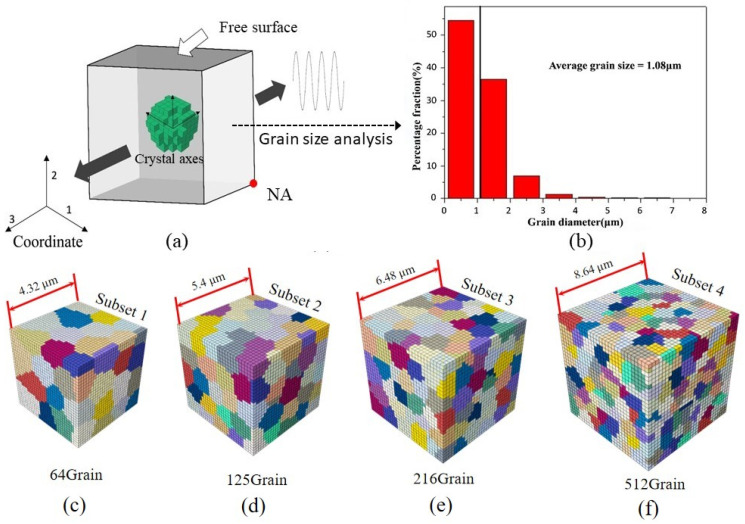
(**a**) Boundary condition and loading direction applied to RVE; (**b**) grain size analysis; (**c**) subset 1–64 grains; (**d**) subset 2–125 grains; (**e**) subset 3–216 grains; (**f**) subset 4–512 grains.

**Figure 5 materials-15-03403-f005:**
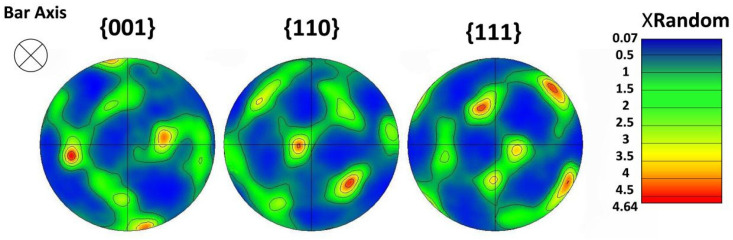
The pole figure of UFG 6061AA.

**Figure 6 materials-15-03403-f006:**
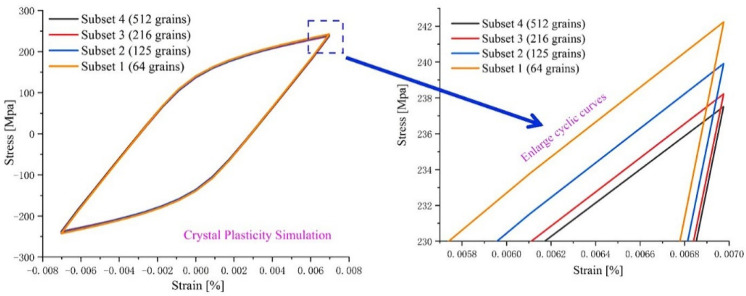
Comparison between the steady hysteresis loops under repeated load and the mechanical response from the polycrystal FE model simulated with different numbers of grains.

**Figure 7 materials-15-03403-f007:**
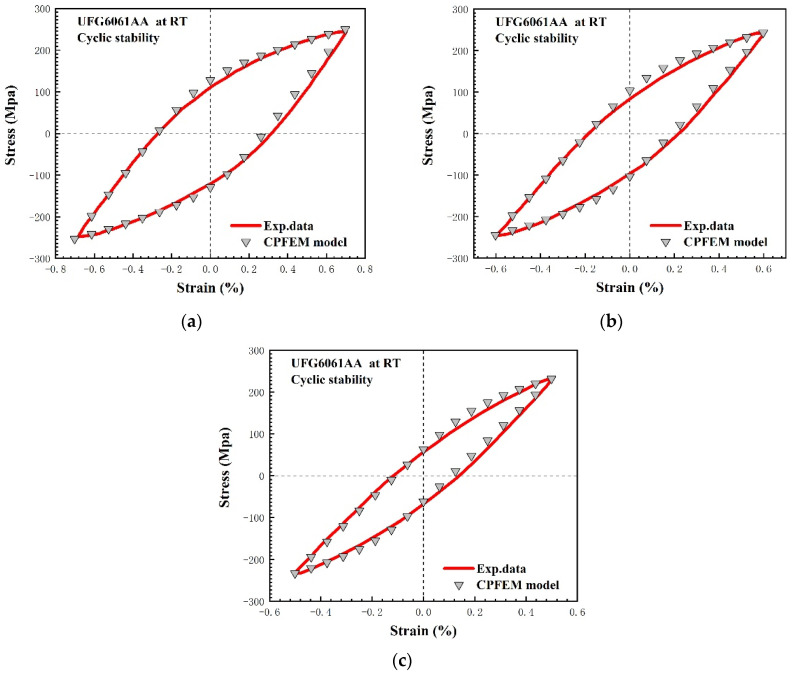
Comparison between the experimental and computational cyclic stress–strain curves in terms of hysteresis loops: (**a**) 0.005; (**b**) 0.006; (**c**) 0.007.

**Figure 8 materials-15-03403-f008:**
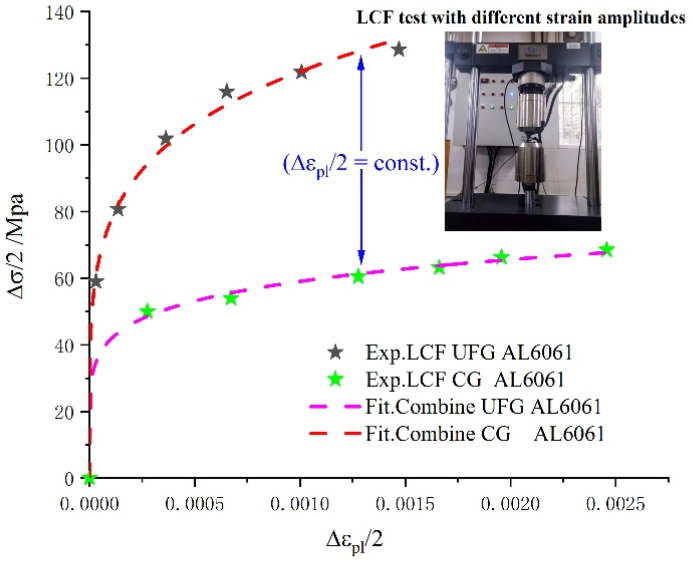
Stress amplitude versus plastic strain amplitude curves of UFG 6061AA.

**Figure 9 materials-15-03403-f009:**
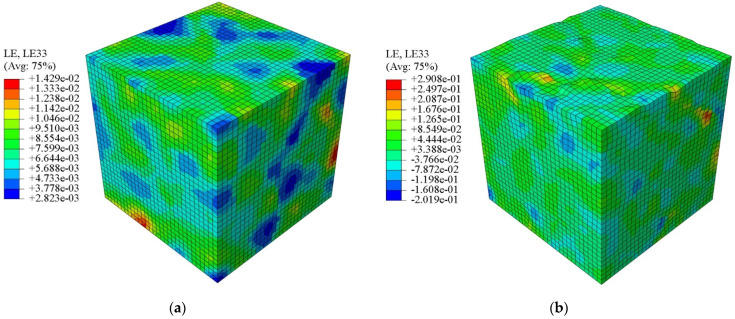
Longitudinal strain contours of the RVE at the maximum tension moment with respect to the (**a**) 3rd cycle and (**b**) 300th cycle under a strain amplitude of 0.7%.

**Figure 10 materials-15-03403-f010:**
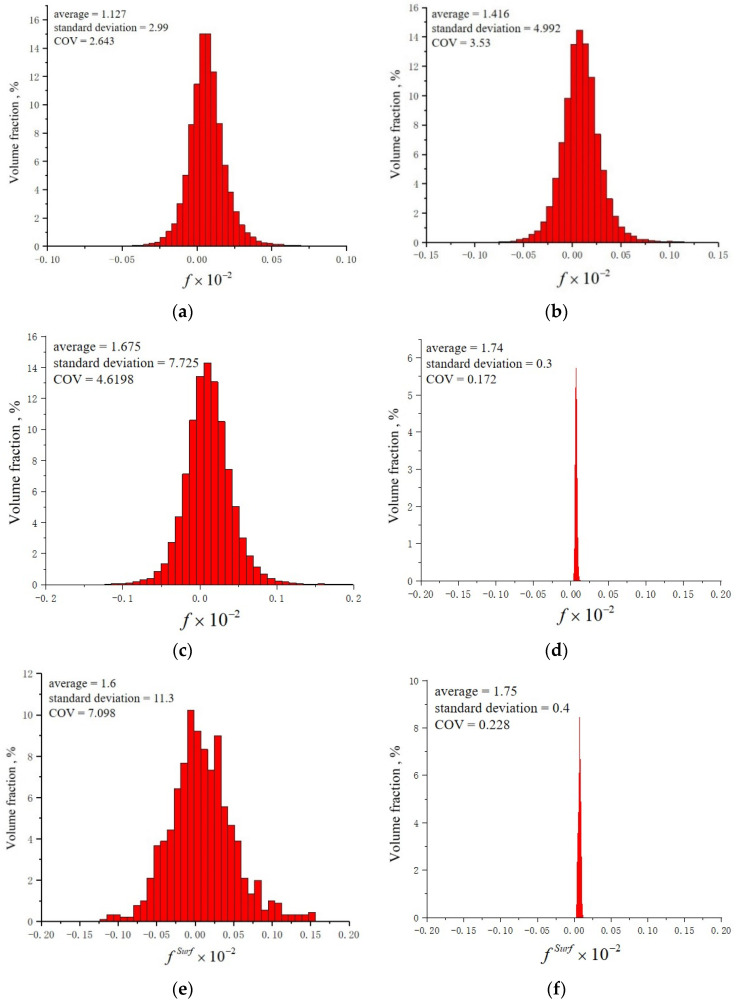
The histograms of the frequency distribution for f at the maximum moment of tension under the strain amplitude of (**a**) 0.5%, (**b**) 0.6%, and (**c**) 0.7% in the 300th cycle. (**d**) The f in the 3rd cycle under the strain amplitude of 0.7%; (**d**) the distribution of $f^{Surf}$ on the free surface in terms of the (**e**) 300th cycle and (**f**) 3rd cycle.

**Figure 11 materials-15-03403-f011:**
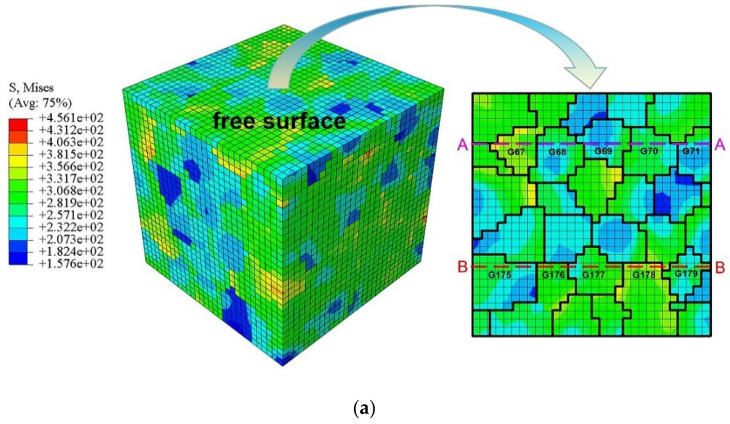
Contours of (**a**) Mises and (**b**) LE maximum principal strain for the whole RVE and its free surface.

**Figure 12 materials-15-03403-f012:**
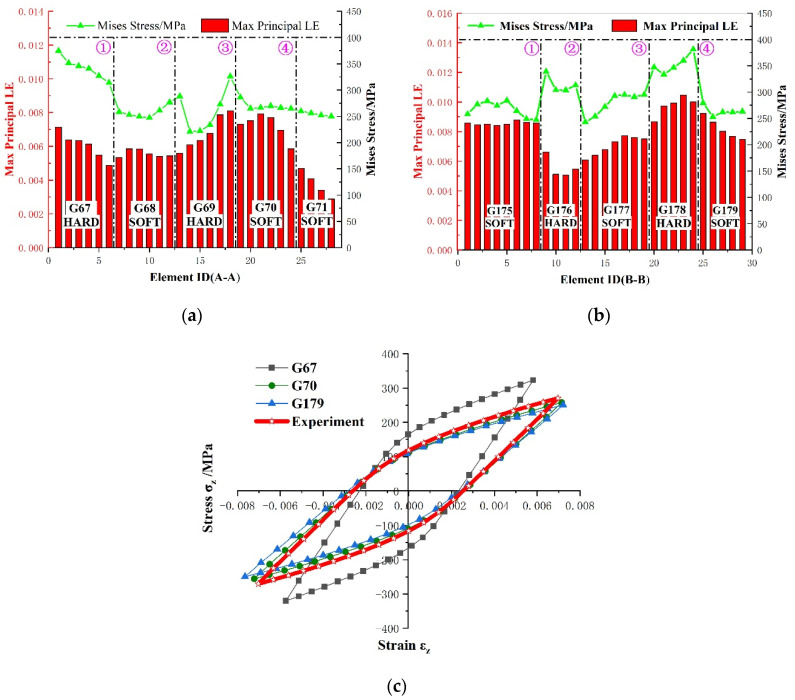
Local stress and strain analysis arising from the preferred crystallographic orientation and neighbor grain constraint effect: along the (**a**) horizontal line A; (**b**) horizontal line B; (**c**) cyclic response for specified grains.

**Figure 13 materials-15-03403-f013:**
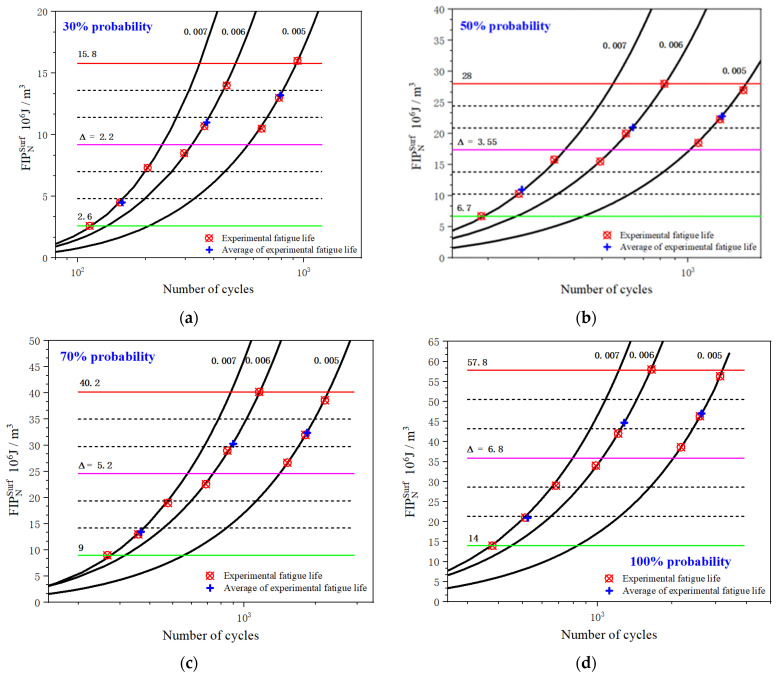
The evolution of $FIP_{N}^{Surf}$ under different probabilities.

**Figure 14 materials-15-03403-f014:**
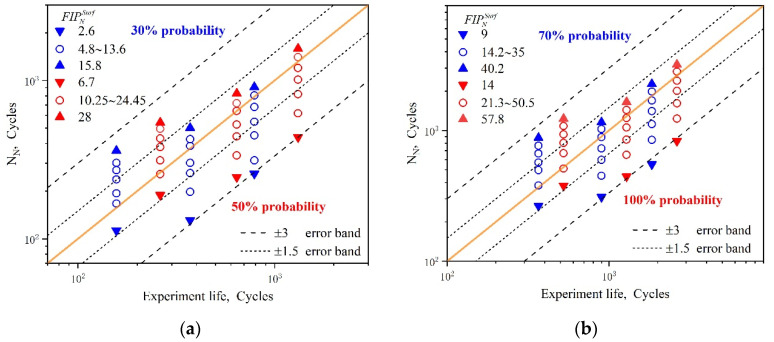
Comparisons between the experimental and predicted loading cycles for nucleation by using $FIP_{N}^{Surf}$ on the free surface: (**a**) 30% probability and 50% probability; (**b**) 70% probability and 100% probability.

**Figure 15 materials-15-03403-f015:**
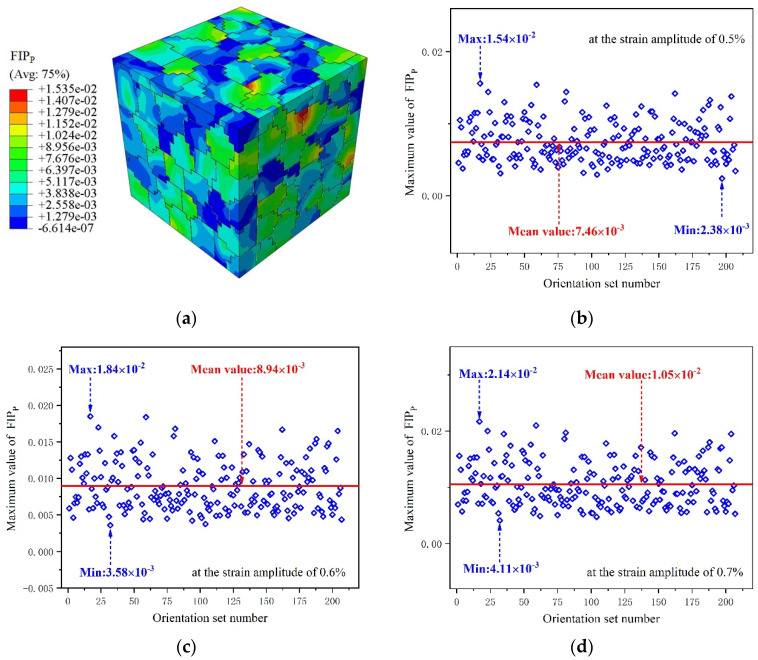
(**a**) Distributions of FIP_p_ by contours with respect to the strain amplitude of 0.5%. Effect of the grain orientation on the maximum FIP_p_ at a strain amplitude of (**b**) 0.5%, (**c**) 0.6%, and (**d**) 0.7%.

**Figure 16 materials-15-03403-f016:**
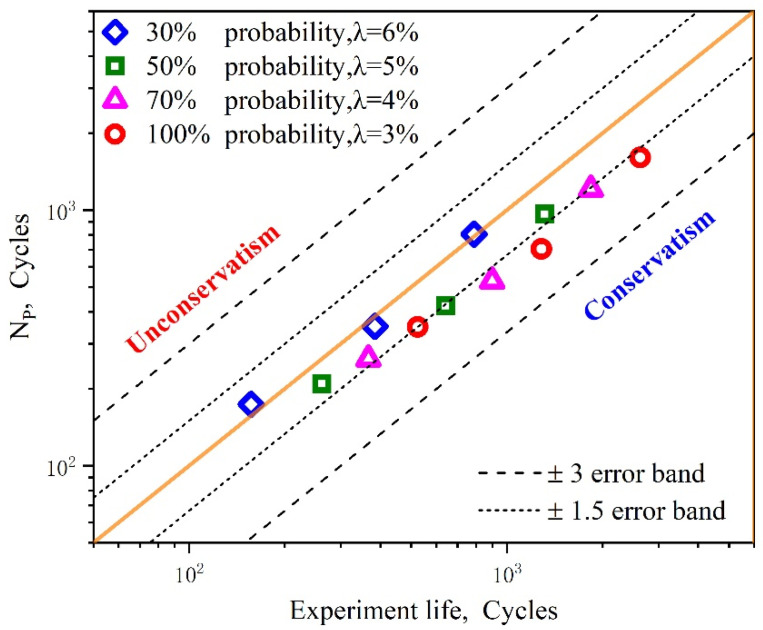
Comparisons between the experimental and predicted small crack propagation life based on the microscale multiaxial FIP_P_ with respect to four probabilities.

**Figure 17 materials-15-03403-f017:**
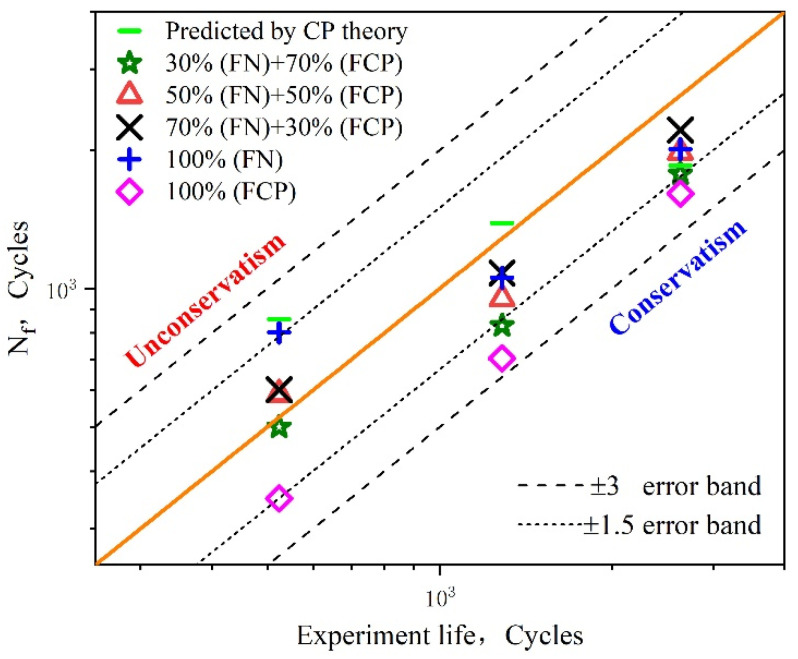
Comparisons between the experiments, predicted results using the proposed method under different probability combinations, and the method based on the crystal plasticity (CP) theory.

**Figure 18 materials-15-03403-f018:**
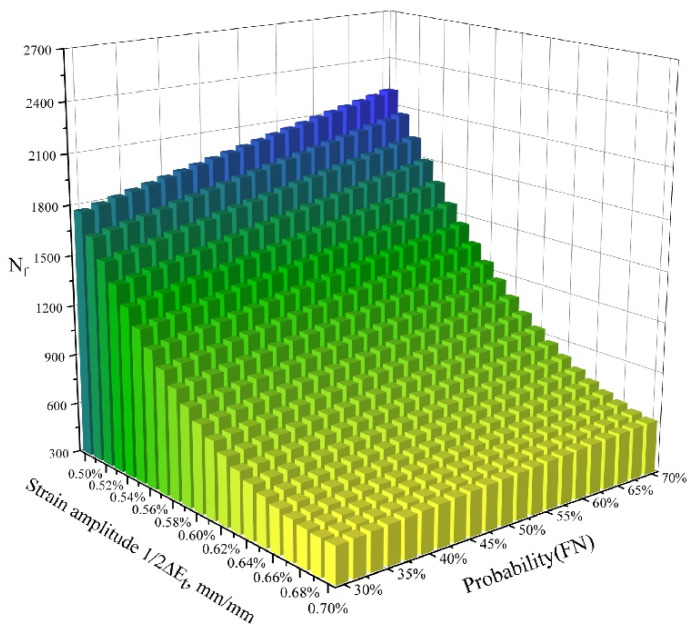
Comparisons between the experiments, predicted results using the proposed method under different probability combinations, and a previously published method based on the crystal plasticity (CP) theory.

**Table 1 materials-15-03403-t001:** Chemical composition of 6061 aluminum alloy (% in weight).

Si	Fe	Cu	Mn	Mg	Cr	Zn	Ti
0.586	0.241	0.264	0.095	0.945	0.07	0.024	0.005

**Table 2 materials-15-03403-t002:** Material parameters for CPFEM simulation.

Elastic Constants and Flow Parameters			Hardening Parameters		
Parameter	Dimension	Value	Parameter	Dimension	Value
C_11_	MPa	77,159	d_mean_	μm	1.08
C_12_	MPa	55,706	G	MPa	26,209
C_44_	MPa	35,910	$\omega_{1}$	-	1.0
$v_{0}$	s^−1^	1013	$\omega_{2}$	-	1.1
$\rho_{0}$	1014 mm^−2^	1.5	$h_{0}$	-	0.014
c	-	0.014	$y_{c}$	-	$0.9\times 10^{-25}$
b	nm	0.286 [47]	$r_{D}$	MPa	118
$\Delta G_{0}$	ev	1.4 [51]	$h_{b}$	MPa	8831
m	-	0.28	$\Delta G_{B}$	J/mol	87,000 [51]
$k_{B}$	J/K	$1.38\times 10^{-23}$	u	-	3.55
p	-	0.99			
q	-	1.01			
T	K	298			

**Table 3 materials-15-03403-t003:** Characteristic parameters of UFG 6061AA and CG 6061AA.

Materials	$n^{\prime }$	${K^{\prime }}_{UFG}$	Cyclic Yield Strength $R_{P,0.2}^{UFG}\;(\mathrm{MPa})$	Strengthening Factor $K_{GBS}$
UFG 6061AA/RT State	0.21	484	139	2.1
CG AA6061/RT State	0.15	168	66	

**Table 4 materials-15-03403-t004:** Nucleation life prediction for various $FIP_{N}^{Surf}$ with respect to their different probabilities.

		**30% Probability**							**50% Probability**						
	$FIP_{N}^{Surf}$	2.6	4.8	7	9.2	11.4	13.6	15.8	6.7	10.25	13.8	17.35	20.9	24.45	28
$\Delta E_{t}/2$															
0.5%		258	313	450	568	680	804	908	438	621	820	1013	1201	1406	1589
0.6%		131	198	260	302	386	426	502	245	336	443	527	642	718	827
0.7%		113	167	194	237	271	302	360	189	256	313	380	430	495	543
		**70% Probability**							**100% Probability**						
	$FIP_{N}^{Surf}$	9	14.2	19.4	24.6	29.8	35	40.2	14	21.3	28.6	35.9	43.2	50.5	57.8
$\Delta E_{t}/2\;$															
0.5%		551	846	1118	1406	1693	1981	2265	830	1223	1615	2010	2400	2818	3185
0.6%		310	450	596	728	889	1026	1157	445	652	851	1054	1245	1436	1653
0.7%		265	380	497	428	667	765	880	378	512	670	802	933	1081	1228

**Table 5 materials-15-03403-t005:** Tested LCF lives of the UFG AA6061 vs. the predicted results considering only one FIP.

$\Delta E_{t}/2\;(\%)$	$N_{f}$	${\bar{N}}_{f}$	$N_{N}$	$\frac{N_{N}\;-{\;\bar{N}}_{f}}{{\;\bar{N}}_{f}}$	$N_{p}$	$\frac{N_{p}\;-\;{\bar{N}}_{f}}{{\;\bar{N}}_{f}}$
0.5	3125/2173/2589	2629	2010	−0.24	1608	−0.39
0.6	1653/986/1213	1284	1054	−0.18	704	−0.45
0.7	512/378/681	524	802	0.53	349	−0.33

**Table 6 materials-15-03403-t006:** Computation results and prediction errors made by N1, N2, N3, and N4 schemes.

N_1_: 30%(FN) + 70%(FCP)		N_2_: 50%(FN) + 50%(FCP)		N_3_: 70%(FN) + 30%(FCP)		N_4_: Previous Published Method	
N_1_	$\frac{N_{1}\;-{\;\bar{N}}_{f}}{{\;\bar{N}}_{f}}$	N_2_	$\frac{N_{2}\;-{\;\bar{N}}_{f}}{{\;\bar{N}}_{f}}$	N_3_	$\frac{N_{3}\;-{\;\bar{N}}_{f}}{{\;\bar{N}}_{f}}$	N_4_	$\frac{N_{4}\;-{\;\bar{N}}_{f}}{{\;\bar{N}}_{f}}$
1774	−0.33	1978	−0.25	2210	−0.16	1853	−0.30
830	−0.35	949	−0.26	1079	−0.16	1385	0.08
499	−0.05	589	0.12	602	0.15	856	0.63

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
