# Peer review of "An Approach for Predicting the Low-Cycle-Fatigue Crack Initiation Life of Ultrafine-Grained Aluminum Alloy Considering Inhomogeneous Deformation and Microscale Multiaxial Strain"

_materials, 2022, doi:10.3390/ma15093403_

Round 1
Reviewer 1 Report
The cyclic loading of the structural parts is one of the most common reasons of the fracturing. The experimental and theoretical investigation of the fatigue cracks initiation is important for prolongation of working life of the metallic components. The authors of the paper theoretically calculated using finite element simulation the distribution of strain inhomogenities during low-cyclic test. The comparison of the calculated results with experimental has shown an appropriate convergence (minimum average error was about 16 %). The presented results seem to be interesting. However, some points of the manuscript are needed to be more clear. The paper should be modified accordingly following comments:
- The constructed model seems to be too simplify. The authors did not consider roughness of the surface in the real material, heating during the fatigue testing, and other factors. Despite of good accordance with the experiment he authors should describe how their models may be applied to the real materials considering the presence of the surface and internal defects.
- It is known, that the main reason of the initiation of the fatigue cracks is the movement of the dislocations to the surface (Cottrell or Mott mechanisms). For example, accordingly Cottrell theory, if two sources of dislocations act near the surface it may give steps which should be as stress concentrators and provide crack initiation. In the Figure 1 the process of the crack propagation through the plastic deformation is shown. However, the authors did not consider in their simulations the influence of the dislocations on the crack initialization.
- In Figure 7b the significant deformation is seen on the free surface. However, the level of the longitudinal strain is same such on the internal surface. It seems to be very strange. How did the authors may describe this fact?
- The shear modulus of Al alloys has a value of about 26000 MPa. Why did the authors use in their calculation the value 18346 MPa (Table 1)?
- The details about the experimental testing (equipment, size of the samples, roughness of the surface, strain rate, etc.) should be added to the manuscript. What was the initial microstructure of fine-grain A6061 alloy?
- Minor corrections:
- It is better to provide the names of parameters in Table 1 together with letters. It is hard to find each parameter in the text.
Reviewer 2 Report
The manuscript explained an approach for predicting the low-cycle fatigue crack initiation life of ultrafine grained aluminium alloy considering the inhomogeneous deformation and microscale multiaxial strain. The manuscript needs a major revision prior considering for publication:
1- The abstract is not comprehensive, and does not reflect the main finding of the paper.
2- The traditional definition of fatigue crack divides the cracks into two stages: Crack initiation and crack propagation. Please use the regular definition and avoid using non-famous terms e.g. MSC. Please also cite the relevant references for this definition and use this in the entire manuscript. https://doi.org/10.1002/srin.202000242, https://doi.org/10.1016/S0142-1123(01)00033-0 Also, please explain the LCF vs HCF by these references. It is usual that in LCF majority of the life spends in the propagation stage, not initiation. All these concepts should be introduced and explained.
3- Generally, the introduction is not concrete and convincing. The introductory part should show us the motivation clearly. e.g. 'In the last decade some researchers tried to interpret the fatigue crack initiation behavior in connection with crystal plasticity theory with the help of FIPs.' Explain the previous findings in detail.
4- How the obtained results of predication can fit the experimental results? You need to validate your results.
5- Fig. 16 needs more explanation. The explanation is ambiguous.
6- Compare the validity and accuracy of the current results with the same method which was used in the previously published papers.
7- The English language of the manuscript is not appropriate for a scientific publication. Therefore, it should be revised by an English native speaker who is an expert in the field.
Reviewer 3 Report
In the paper, the Authors presented valuable fatigue crack initiation life analysis results. The aluminium alloy considering the inhomogeneous deformation and microscale multiaxial strain was investigated. The main findings can be expressed as "Regarding the two newly proposed FIPs as the driving force for the nucleation and MSC propagation, respectively, we have predicted the nucleation life and MSC propagation life with respect to different probability factors. The predicted accuracy of the fatigue crack initiation life based on only one FIP was acceptable, when the error bands were set to the range of ±3.". As a Reviewer, I would like to express my positive opinion. I would like to recommend accept after minor revision (corrections to minor methodological errors and text editing) decision.
Authors should describe in a complex way, methodology, planned fatigue experiments are well correlated with new parameters. Mentioned below aspects must be taken into consideration during the revision:
Nomenclature:
(1) I suggest adding "Nomenclature" section (with units and abbreviations) in the manuscript.
Simulation methodology:
(2) The authors should present the new Figure with an algorithm of all calculation steps, for example in the form of a flowchart. This will allow the reader to quickly understand the method.
Material:
(3) Please provide information (taken from literature or by Authors investigation) about chemical composition and mechanical parameters for tested materials. How were the mechanical parameters (Table 1) obtained? From literature or determined by the authors for the tested material? Not clear.
Experimental tests:
(4) Please provide more information about testing machine and software.
(5) Did the specimen preparation and test conduct with any standard? Which?
(6) Please include the Figure showing the shape and dimensions of the tested specimens.
(7) Please provide information on how many samples were tested.
Results:
(8) The main limitations of the present method must be identified and discussed in the end of this section.
References:
(9) References/Introduction section should be extended. I propose to add a few entries in the Introduction section, regarding the crack initiation and fracture surface investigation, especially published in Materials journal [1,2].
- Tang, Z.; Chen, Z.; He, Z.; Hu, X.; Xue, H.; Zhuge, H. Experimental and Numerical Study of Combined High and Low Cycle Fatigue Performance of Low Alloy Steel and Engineering Application. Materials 2021, Vol. 14, Page 3395 2021, 14, 3395, doi:10.3390/MA14123395.
- Macek, W.; Branco, R.; Costa, J.D.; Trembacz, J. Fracture Surface Behavior of 34CrNiMo6 High-Strength Steel Bars with Blind Holes under Bending-Torsion Fatigue. Materials 2022, Vol. 15, Page 80 2021, 15, 80, doi:10.3390/MA15010080.
Round 2
Reviewer 1 Report
The authors have answered previous comments and improved the manuscript. The paper may be accepted for publication.
Reviewer 2 Report
The revised version seems better than the previous verison, and it suits for publcation. Some format and style checking are needed before publication.